# Association between grip strength and anthropometric characteristics in the community-dwelling elderly population in Taiwan

Ming-Hsun Lin[1], Chun-Yung Chang[1,2], Chieh-Hua Lu[1], Der-Min Wu[3], Feng-Chih Kuo[1], Che-Chun Kuo[4], Nain-Feng Chu [1,3]*

1 Division of Endocrinology and Metabolism, Department of Internal Medicine, Tri-Service General Hospital, National Defense Medical Center, Taipei, Taiwan, Republic of China, 2 Department of Internal Medicine, Kaohsiung Armed Forces General Hospital, Kaohsiung City, Taiwan, Republic of China, 3 School of Public Health, National Defense Medical Center, Taipei, Taiwan, Republic of China, 4 Department of Internal Medicine, Taoyuan Armed Forces General Hospital, Taiwan, Republic of China

* chuepi369@gmail.com, chuepi@mail.ndmctsgh.edu.tw

**Data Availability Statement:** All relevant data are within the manuscript and its Supporting Information files.

## Abstract

### Background

Sarcopenia and muscle weakness in elderly are contributed burden of public health and impact on quality of life. Weak grip strength was key role in diagnosis of sarcopenia and reported increased mortality, function declined in elderly. This study evaluated the association between GS and each common anthropometric characteristic in community-dwelling elderly.

### Design and method

From 2017 to 2019, we conducted a community-based health survey among the elderly in Chiayi county, Taiwan. Participants were 65 years old or older, and total of 3,739 elderly subjects (1,600 males and 2,139 females) with a mean age of 76 years (range 65–85 years old) were recruited. General demographic data and lifestyle patterns were measured using a standard questionnaire. Anthropometric characteristics such as body height, body weight, body mass index (BMI), body waist and hip circumference, and body fat were measured by standard methods. GS was measured using a digital dynamometers (TKK5101) method.

### Results

The mean GS was 32.8 ± 7.1 kg for males and 21.6 ± 4.8 kg for females (p < 0.001). For both sexes, elderly subjects with the same body weight but smaller body waist circumference had greater GS. The subjects with the same body waist size but heavier weight had greater GS. Furthermore, after adjusting for age, lifestyles, disease status, and potential anthropometric variable, multivariate regression analyses indicated that BMI was positively associated with GS (for males, beta = 0.310 and for females beta = 0.143, both p < 0.001)

**Funding:** The authors received no specific funding for this work.

**Competing interests:** The authors have declared that no competing interests exist.

and body waist was negatively associated with GS (for males, beta = −0.108, p < 0.001; for females, beta = −0.030, p = 0.061).

## Conclusions

This study suggested that old adults with higher waist circumstance had weaker GS. Waist circumstance was negatively associated with GS, body weight was positively associated with GS in contrast. It may implies that central obesity was more important than overweight for GS in elderly.

## Introduction

Global ageing is a phenomenon because of declining fertility and increasing life expectancies [1]. Sarcopenia is an age-related pathophysiological process of skeletal muscle loss and muscle strength [2]. The prevalence of sarcopenia is from 6.9% to 63% among different populations and countries [3]. Sarcopenia and muscle weakness are risk factors for physical disability, falls and mortality [4, 5]. Thus, it is not only an important public health issue but also a clinical issue among the elderly. Several causes of sarcopenia include reduction of testosterone and estrogen due to age [6], decline of physical activity [7] and increased insulin resistance with ageing [8].

Grip strength (GS) and gait speed are measures that detect muscle function and are diagnostic criteria for sarcopenia according to the definition of the European Working Group on Sarcopenia in Older People [9]. Nevertheless, GS is a more common, effective, quick and easy method for evaluating the muscle strength [10, 11]. It is also a good predictor of function and disability among the elderly. Good GS is a protective factor for frailty and disability among elderly population [12].

Anthropometric parameters, such as body height (BH), body weight (BW), body waist circumference (WC), body hip circumference (HIP), body fat and body mass index (BMI) are easy methods to evaluate body composition. There have been several studies that reported the relationship between anthropometric parameters and GS among Asian, European and American populations [13–17]. Body waist is proportional to central adiposity, which is associated with insulin resistance, morbidity and mortality. A previous report suggested that the prevalence of sarcopenia among the elderly was lower among those with waist circumference-defined abdominal obesity than those without abdominal obesity [18]. But the central obesity was proven to negatively associate with sarcopenia recently [19]. Hence, the association between GS and body waist in older adults are not clear.

As a rapid ageing society in Taiwan, few studies have investigated the association between GS and anthropometric parameters [20, 21]. This study aimed to demonstrate the epidemiological characteristics of associations between in the GS and each common anthropometric characteristic in community-dwelling older people. Moreover, it examined that whether GS was negatively corrected central obesity and which anthropometric characteristic is potentially the most important correlated to GS.

## Materials and methods

### Study population

From 2017 to 2019, we conducted a series of community-based health surveys of the middle-aged and elderly populations in Chiayi, Taiwan. People aged 65 years or older and lived in

Chiayi county were invited to participate in the survey. A survey has been conducted in Chiayi county every 3 years. A total of 3,739 elderly subjects (1,600 males and 2,139 females) with a mean age of 76 years (range 65–85 years old) participated in the study (S2 File). The inclusion criteria for this study were elderly, aged 65–85 years, and without infection or acute disorders within the previous three weeks.

## Questionnaire

General demographic data and lifestyle patterns (including dietary pattern, habit of smoking and alcohol intake) were measured using a standard structured questionnaire (as S1 File). Disease status, such as cardiovascular disease (CVD), cerebrovascular disease (CVA), hypertension, dyslipidemia, diabetes mellitus (DM), chronic kidney disease (CKD), and current medications, was also recorded from the study population.

## Anthropometric measurements

Anthropometric characteristics such as BH, BW, WC, HIP and body fat were measured using standard methods. We had trained our staffs before survey conduction. All anthropometric measures, including BH, BW, WC, HIP and body fat, reach the inter-observer variation less than 5% and intra-observer variation around 3–5%. Participants were barefoot and wore light indoor clothing. BH was recorded to the nearest 0.5 cm using a stadiometer. BW was measured to an accuracy of 0.1 kg using a standard beam balance scale (TBF-410, Tanita Corp., Tokyo, Japan). Body fat was measured using a segmental body composition analyser (TBF-410, Tanita Corp., Tokyo, Japan). WC was measured to the nearest 0.1 cm at the midpoint between the margin of the last rib and the iliac crest of the ilium. HIP was measured at the widest part of the pelvic region. We calculated the BMI as BW (kg) divided by the square of height ($m^2$) and calculated the waist-to-hip ratio (WHR) as WC (cm) divided by the HIP (cm).

## GS measurement

GS was measured using a digital dynamometers (TKK5101) method, which is a tool with an adjustable grip span, ranging from 3.5 to 7 cm and weighing from 5 to 100 kg with minimal difference around 0.1 kg [22]. All the participants were in a sitting position with fully extended elbows [23]. Then, we measured GS on the dominant hand after 2–3 minutes of resting. Two GS measurements were recorded, and the mean value was used for analyses. For grip strength, the inter-observer variation was less than 5% and intra-observer variation was around 3%, and the standard error mean was about 0.2 ~ 0.3.

The definition of normal GS according to European Working Group on Sarcopenia in Older People (EWGSOP) was ≥ 30kg in the male and ≥ 20 kg in the female. Weak GS was defined as GS < 30 kg in the male and GS < 20 kg in the female [9].

## Approval of the IRB

All participants provided written informed consent and agreed to provide their general demographic data, questionnaire answers, anthropometric data and blood samples for the study. The institutional review board of Tri-service General Hospital approved the study (Number: TSGHIRB-1-108-05-073).

## Statistical methods

We used SPSS ver-22 (IBM Corporation, New York, NY) to conduct all statistical analyses. We analysed the sample means and SDs of continuous variables, such as anthropometric measures

and GS. The Mann–Whitney U test was used to compare the differences between groups. The Kruskal–Wallis H test and post hoc test were used for comparisons of subgroups and to compare more than three groups. Categorical variables were described by number and percentages. A Chi-squared test was used to compare the differences among two or more groups. Spearman's rank correlation coefficient was used to compare variables. We used multivariate regression analyses to examine the association between anthropometrics variables and grip strength. A two-tailed *p* value of less than 0.05 was considered statistically significant.

## Results

In this present study, Tables 1 and 2 shows the general characteristics of all participants with gender specifications. Male had heavier body weight (65.8 ± 10.1 kg in males and 57.1 ± 9.4 kg in females, *p* < 0.001) and larger body waist size (88.4 ± 9.1 cm in males and 83.1 ± 9.3 cm in females, respectively, *p* < 0.001). The body hip was similar between males and females (95.0 ± 6.5 cm in males and 95.6 ± 7.6 cm in females, *p* > 0.05). Other anthropometric data, including body height and WHR, were significantly greater in males. However, the BMI (kg/m$^2$) was similar between both sexes (24.9 ± 3.4 in males and 25.1 ± 3.8 in females, *p* >0.05). Moreover, females also had higher body fat than males (22.7 ± 6.4 in males and 32.4 ± 7.4 in females, *p* < 0.001). The GS was higher in males; the mean grip strength was 32.8 ± 7.1 kg for

**Table 1. General characteristics and grip strength among male elderly population (n = 1,600).**

| Variables | Grips Strength | | | | | | p-value |
|---|---|---|---|---|---|---|---|
| | Normal(GS ≥30 kg) | | | Weak (GS <30 kg) | | | |
| | (n = 1,061) | | | (n = 539) | | | |
| | Mean | ± | SD | Mean | ± | SD | |
| Age (years)[†] | 71.5±5.2 | | | 76.7±6.1 | | | <0.001*** |
| Body height (cm) | 163.8±5.7 | | | 159.6±5.8 | | | <0.001*** |
| Body weight (kg) | 67.7±9.8 | | | 61.9±9.7 | | | <0.001*** |
| BMI (kg/m$^2$) | 25.2±3.3 | | | 24.3±3.5 | | | <0.001*** |
| Body waist (cm) | 88.9±8.9 | | | 87.3±9.3 | | | 0.001** |
| Body hip (cm) | 95.76.3 | | | 93.7±6.6 | | | <0.001*** |
| WHR | 0.9±0.1 | | | 0.9±0.1 | | | 0.465 |
| Body fat (%) | 23.0±6.2 | | | 22.0±6.8 | | | 0.003** |
| Grips strength (kg) | 36.7±5.0 | | | 25.2±3.6 | | | <0.001*** |
| Chronic disease[‡] | (n) | | (%) | (n) | | (%) | |
| CVD | 157 | | 14.8 | 95 | | 17.6 | 0.142 |
| CVA | 30 | | 2.8 | 32 | | 5.9 | 0.002** |
| Hypertension | 452 | | 42.6 | 239 | | 44.3 | 0.507 |
| Dyslipidemia | 156 | | 14.7 | 63 | | 11.7 | 0.097 |
| DM | 218 | | 20.5 | 116 | | 21.5 | 0.650 |
| CKD | 36 | | 3.4 | 29 | | 5.4 | 0.057 |
| Behavior status | (n) | | (%) | (n) | | (%) | |
| Smoking | 143 | | 13.5 | 70 | | 13.0 | 0.785 |
| Alcohol drinking | 199 | | 18.8 | 78 | | 14.5 | 0.032* |

Abbreviations: BMI, Body mass index; WHR: Body waist to hip ratio; CVD, Cardiovascular disease; CVA, Cerebrovascular disease; DM, Diabetes mellitus; CKD, Chronic kidney disease.

[†] t test was compared with grip strength normal and grip strength weak among characteristics of anthropometry and grip strength; ***p<0.001, **p<0.01, *p<0.05.

[‡] chi-square test was compared with grip strength normal and grip strength weak among characteristics of behavior status and chronic diseases; ***p<0.001, **p<0.01, *p<0.05.

**Table 2. General characteristics and grip strength among female elderly population (n = 2,139).**

| Variables | Grips Strength | | | | | | p-value |
|---|---|---|---|---|---|---|---|
| | Normal (GS ≥30 kg) | | | Weak (GS <20 kg) | | | |
| | (n = 1,377) | | | (n = 762) | | | |
| | Mean | ± | SD | Mean | ± | SD | |
| Age (years)[†] | 71.3±5.3 | | | 75.2±6.2 | | | <0.001*** |
| Body height (cm) | 152.0±5.4 | | | 148.7±5.6 | | | <0.001*** |
| Body weight (kg) | 58.5±9.2 | | | 54.6±9.4 | | | <0.001*** |
| BMI(kg/m²) | 25.3±3.8 | | | 24.7±4.0 | | | <0.001*** |
| Body waist (cm) | 83.4±9.2 | | | 82.7±9.5 | | | 0.133 |
| Body hip (cm) | 96.2±7.2 | | | 94.6±8.1 | | | <0.001*** |
| WHR | 0.9±0.1 | | | 0.9±0.1 | | | 0.004** |
| Body fat (%) | 32.9±7.2 | | | 31.4±7.6 | | | <0.001*** |
| Grips strength (kg) | 24.4±3.3 | | | 16.6±2.5 | | | <0.001*** |
| Chronic disease[‡] | (n) | (%) | | (n) | (%) | | |
| CVD | 190 | 13.8 | | 132 | 17.3 | | 0.029* |
| CVA | 25 | 1.8 | | 11 | 1.4 | | 0.522 |
| Hypertension | 593 | 43.1 | | 358 | 47.0 | | 0.081 |
| Dyslipidemia | 220 | 16.0 | | 117 | 15.4 | | 0.705 |
| DM | 255 | 18.5 | | 168 | 22.0 | | 0.050* |
| CKD | 34 | 2.5 | | 18 | 2.4 | | 0.878 |
| Behavior status | (n) | (%) | | (n) | (%) | | |
| Smoking | 13 | 0.9 | | 6 | 0.8 | | 0.711 |
| Alcohol drinking | 38 | 2.8 | | 9 | 1.2 | | 0.017* |

Abbreviations: BMI, Body mass index; WHR: Body waist to hip ratio; CVD, Cardiovascular disease; CVA, Cerebrovascular disease; DM, Diabetes mellitus; CKD, Chronic kidney disease.

[†] t test was compared with grip strength normal and grip strength weak among characteristics of anthropometry and grip strength; ***p<0.001, **p<0.01, *p<0.05.

[‡] chi-square test was compared with grip strength normal and grip strength weak among characteristics of behavior status and chronic diseases; ***p<0.001, **p<0.01, *p<0.05.

males and 21.6 ± 4.8 kg for females ($p < 0.001$). The GS showed significantly differently that the elderly with CVA had weak grip strength in male, but no difference in female ($p = 0.002$ in male and $p = 0.522$ in female, respectively). In contrast, the GS showed significantly differently that the elderly with DM had weak grip strength in female, but not in male ($p = 0.650$ in male and $p = 0.05$ in female, respectively). The elderly with CVD also demonstrated weaken GS in both genders, although no significant in male ($p = 0.142$ in male and $p = 0.029$ in female, respectively). And the elderly with behaviour of alcohol drinking showed significant higher GS in both genders ($p = 0.032$ in male and $p = 0.017$ in female, respectively).

Table 3 shows Spearman's correlation coefficients between GS and anthropometric variables. Age was negatively correlated to GS in both sexes. BH, BW, BMI, WC, HIP and body fat were all significantly positively correlated with GS ($p < 0.001$) in both sexes. Compared with other anthropometric measures, only WHR showed a negative correlation but the difference was only statistical significance in female ($r = -0.013$, $p = 0.613$ in males and $r = -0.047$, $p = 0.030$ in females).

Table 4 summarises the distribution of GS among the tertile subgroups of WC and BW in both sexes. We divided WC results into three subgroups; i.e. smallest body waist (WC1), moderate body waist (WC2) and largest body waist (WC3). The BW results were also divided into three groups: lowest body weight (BW1), moderate body weight (BW2) and highest body

**Table 3. Spearman correlation between grip strength and anthropometric variables among elderly population with gender specifications.**

| | Male(n = 1,600) | | Female(n = 2,139) | |
|---|---|---|---|---|
| | coefficient | p-value | coefficient | p-value |
| Age (years) | -0.458 | <0.001*** | -0.364 | <0.001*** |
| Body height (cm) | 0.415 | <0.001*** | 0.348 | <0.001*** |
| Body weight (kg) | 0.345 | <0.001*** | 0.284 | <0.001*** |
| BMI (kg/m²) | 0.177 | <0.001*** | 0.138 | <0.001*** |
| Body waist (cm) | 0.130 | <0.001*** | 0.083 | <0.001*** |
| Body hip (cm) | 0.218 | <0.001*** | 0.178 | <0.001*** |
| WHR | -0.013 | 0.613 | -0.047 | 0.030* |
| Body fat (%) | 0.111 | <0.001*** | 0.152 | <0.001*** |

Abbreviations: BMI, Body mass index; WHR: Body waist to hip ratio.

Spearman correlation was used for the association between grip strength and anthropometric variables ***p<0.001, **p<0.01

*p<0.05.

weight (BW3). After bivariate analyses, the highest GS was found in the elderly with the highest BW and smallest WC in both sexes [a Kruskal–Wallis H test and post hoc test revealed a significant difference ($p < 0.05$) for different groups in both sexes]. The lowest GS was found in the elderly with the lowest BW and largest WC [a Kruskal–Wallis H test and post hoc test revealed a significant difference ($p < 0.05$) for different groups in both sexes].

Table 5 shows the results of multivariate regression analyses for anthropometric variables and GS. In Model I, after adjusting for age, lifestyles, and disease status (CVD, CVA, hypertension, dyslipidemia, DM, and CKD), the regression coefficient and standard error showed a positive result for all anthropometric variables (except WHR). However, after adjusting for

**Table 4. Grip strength distribution (Mean ± SD) among elderly population with classification by body weight and body waist with gender specification.**

| Variables | Body waist (cm) | | | ANOVA† | Post Hoc Test |
|---|---|---|---|---|---|
| | WC1 | WC2 | WC3 | | |
| Male | <84.5 cm | 84.5–92.0 cm | >92 cm | | |
| (n = 1,600)ᐞ | (n = 531) | (n = 535) | (n = 534) | | |
| WT1 (n = 534) | 30.3±6.3 | 28.7±6.2 | 29.0±6.6 | F = 3.49* | T1>T2 |
| WT2 (n = 534) | 35.2±6.6 | 32.7±6.2 | 31.3±6.8 | F = 12.24*** | T1>T2,T1>T3 |
| WT3 (n = 532) | 38.6±7.1 | 37.4±7.7 | 34.9±7.2 | F = 6.38** | T1>T2,T2>T3 |
| Female | <79 cm | 79–87 cm | >87 cm | | |
| (n = 2139)★ | (n = 698) | (n = 711) | (n = 730) | | |
| WT1 (n = 716) | 20.1±4.3 | 19.9±4.3 | 18.4±3.6 | F = 4.24* | T1>T3 |
| WT2 (n = 718) | 22.7±4.3 | 21.8±4.2 | 20.8±5.0 | F = 8.01*** | T1>T3,T2>T3 |
| WT3 (n = 705) | 24.2±4.4 | 23.6±5.4 | 22.9±5.0 | F = 1.63 | |

Abbreviations: WT1, body weight (kg) tertile 1 (lowest); WT2, body weight (kg) tertile 2; WT3, body weight (kg) tertile 3 (highest); WC1, body waist (cm) tertile 1 (lowest); WC2: body waist (cm) tertile 2; WC3, body waist (cm) tertile 3 (highest). T1, body waist (cm) tertile 1 (lowest);T2: body waist (cm) tertile 2; T3, body waist(cm) tertile 3(highest).

† ANOVA F test was to compare these three body waist tertile subgroups of population in grip strength among each body weight tertile specifications

***p<0.001

**p<0.01

*p<0.05.

ᐞThe cut-off values were 61.2 kg between BW1 and BW2, and 69.1 kg between BW2 and BW3 in male.

★The cut-off values were 52.7 kg between BW1 and BW2, and 60.6 kg between BW2 and BW3 in female.

**Table 5. Multivariate regression analysis for anthropometric variables on grip strength with gender specifications.**

| Independent variables | Model I[†] | | | Model II[‡] | | |
|---|---|---|---|---|---|---|
| | β | se β | p-value | β | se β | p-value |
| Male (n = 1,600) | | | | | | |
| Body height (cm) | 0.408 | 0.025 | <0.001*** | 0.375 | 0.026 | <0.001*** |
| Body weight (kg) | 0.198 | 0.015 | <0.001*** | 0.365 | 0.030 | <0.001*** |
| BMI (kg/m²) | 0.246 | 0.048 | <0.001*** | 0.310 | 0.055 | <0.001*** |
| Body waist (cm) | 0.106 | 0.018 | <0.001*** | -0.108 | 0.031 | <0.001*** |
| Body hip (cm) | 0.207 | 0.024 | <0.001*** | -0.041 | 0.040 | 0.296 |
| WHR | 1.463 | 2.724 | 0.666 | -7.489 | 2.854 | 0.009** |
| Body fat (%) | 0.039 | 0.025 | 0.098 | -0.089 | 0.035 | 0.011* |
| Female (n = 2,139) | | | | | | |
| Body height (cm) | 0.236 | 0.017 | <0.001*** | 0.225 | 0.017 | <0.001*** |
| Body weight (kg) | 0.121 | 0.010 | <0.001*** | 0.219 | 0.020 | <0.001*** |
| BMI (kg/m²) | 0.136 | 0.026 | <0.001*** | 0.143 | 0.027 | <0.001*** |
| Body waist (cm) | 0.055 | 0.011 | <0.001*** | -0.030 | 0.016 | 0.061 |
| Body hip (cm) | 0.086 | 0.013 | <0.001*** | -0.044 | 0.023 | 0.056 |
| WHR | 0.975 | 1.530 | 0.521 | -1.085 | 1.511 | 0.473 |
| Body fat (%) | 0.061 | 0.013 | <0.001*** | 0.016 | 0.022 | 0.479 |

Abbreviations: β, regression coefficient; se, standard error; BMI, Body mass index; WHR: Body waist to hip ratio.

[†] Model I: Adjusting for age, smoking, alcohol drinking, and chronic diseases status (cardiovascular disease, cerebrovascular disease, hypertension, dyslipidemia, diabetes mellitus, and chronic kidney disease).

[‡] Model II: For body height, body weight, and BMI further adjusting for body waist and body hip; for body waist, body hip and WHR further adjusting for body height and body weight; for body fat further adjusting for BMI and WHR.

Multivariate regression analysis for anthropometric variables on grip strength

***p<0.001

**p<0.01

*p<0.05.

potential anthropometric variables, the body waist was negatively associated with GS in both sexes (the coefficient was −0.108 with $p < 0.001$ in males and −0.030 with $p = 0.061$ in females).

## Discussion

In the current ageing society, sarcopenia and physical disability among the elderly population is important that contributed to burden of public health and impact on quality of life [24]. Increasing evidence has shown that there is an increased risk of mortality in individuals with lower GS [18, 25, 26], possible due to cardiovascular and respiratory diseases and cancer [27]. Moreover, lower GS was also shown to be associated with certain non-communicable diseases, such as diabetes [28] or non-alcoholic fatty liver disease [29]. GS seems to be an indispensable biomarker for elderly [30]. Although both of GS and gait speed are key roles to stand for sarcopenia, measurement of GS was relatively easier and safer than gait speed in old adults.

This study was the first large community-observed prospective study to investigate the relationship between anthropometric characteristics and GS among individuals older than 65 years in Taiwan. The GS of elderly was 32.8 ± 7.1 kg in males and 21.6 ± 4.8 kg in females, which was similar to other studies in Asian populations [21, 31, 32]. Ethnic differences were found in GS, and GS was higher in a Western population when compared to an Asian population [33–37]. Not surprisingly, the GS in elderly with CVA had weak grip strength in both sex,

but only showed statically significant difference in male. This is possible because poor stroke outcome of activity limitation assessed from the modified Rankin Scale in female compared with male in previous study [38, 39]. A healthcare participant bias might exist in this study that the females with severe CVA cannot attend to our investigation because of bedridden. Interestingly, the elderly with the behaviour of alcohol drinking had the higher GS, and it was consistent with previous published study [40–42]. Although the alcohol use may weaken and waste skeletal muscle [43], but the mechanism for protective factor of GS in elderly with drinking was still unclear [42].

There were some limitations in our community survey study. There was existing bias of false negatives when using questionnaires and questioning for underlying disease. Moreover, participant bias should be considered if those with a disability could not participate in the survey. Hence, we may have overestimated GS. However, our results indicate an association between anthropometric variables and GS among elderly are still reliable. Unlike results reported in other studies, Lee et al. [32] and Günther et al. [34] found that the GS was correlated with height in both sexes, and weight was correlated in males but not in females. Our data showed that GS was significantly correlated with BH, BW and BMI even after adjusting for age. This finding is consistent with the study of Silventoinen et al. [35], although the population in their study was composed of youth. In summary, previous studies produced similar results indicating that GS is inversely correlated with age but showed a positive correlation with height.

Previous study investigated usually demonstrated the univariate association between GS of each anthropometric variable, this study uses WC and BW as variable factor to perform bivariate analyses of GS (as Table 3) in a large community population. Interestingly, we found that elderly subjects with the same BW but smaller WC had greater GS in both sexes with statistical significance even after ANOVA and post hoc analysis. Conversely, elderly subjects with the same WC but heavier weight had greater GS in both sexes. In order to understand which anthropometric file was possibly the most negatively associated with GS, we use multivariate regression with adjusting for the potential factors (as model 2 in Table 4). It reported that GS was only negatively associated with markers of central obesity (Body waist, body hp and WHR), but positively associated body weight. This result was consistent with our previous study that being adequate overweight might be a protective factor in elderly [44]. Compared with overweight, central obesity was harmful for elderly.

Although the importance of GS is unclear in a clinical scenario, GS represents the nutritional profile [45] and physical obesity, and increasing evidence has shown that there is an increased risk of mortality in individuals with lower GS. In the future, a larger cohort study or clinical trial is required to investigate and support the association between waist circumference and grip strength. GS may be an essential parameter not only to evaluate multiple risk factors, such as cardiometabolic or physical disability in clinical scenarios, but also correlated with age and many anthropometric characteristics.

## Conclusion

To conclude, our study showed that GS in the elderly Taiwanese population was similar to reports on GS in other Asian groups and was weaker than the GS of Western populations. Using bivariate analysis, we found that GS was lower among those with a larger WC in subjects with the same BW. Moreover, the GS of the elderly was highly correlated with BH, BW and BMI but was inversely associated with waist circumference in both sexes. In other words, elderly subjects with central obese had a weaker GS, which is a crucial factor when predicting muscle weakness among the community-dwelling elderly population in Taiwan.

## Supporting information

**S1 File. Chiayi-questionnaire-Eng.** Questionnaire using in the survey.
(PDF)

**S2 File. All-data-eng.** Basic characteristics and anthropometric data, history of all participants.
(XLS)

## Acknowledgments

We thank Ms. Winnie for her English writing and correction. We also thank the staffs of the Chiayi Health Bureau, Ms Yu-Chen Lin and Ms. Chun-Yin Liu, for their technical assistances.

## Author Contributions

**Conceptualization:** Nain-Feng Chu.

**Data curation:** Ming-Hsun Lin, Chun-Yung Chang, Der-Min Wu, Che-Chun Kuo.

**Formal analysis:** Nain-Feng Chu.

**Investigation:** Chun-Yung Chang.

**Methodology:** Der-Min Wu.

**Project administration:** Feng-Chih Kuo.

**Resources:** Chun-Yung Chang, Feng-Chih Kuo, Che-Chun Kuo.

**Supervision:** Chieh-Hua Lu, Der-Min Wu, Nain-Feng Chu.

**Validation:** Chieh-Hua Lu.

**Writing – original draft:** Ming-Hsun Lin.

**Writing – review & editing:** Der-Min Wu, Nain-Feng Chu.

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
