## [Decision Letter · Decision Letter 0]

8 Sep 2021

PONE-D-21-24416Association between anthropometric characteristics and grip strength among elderly population in TaiwanPLOS ONE

Dear Dr. CHU,

Thank you for submitting your manuscript to PLOS ONE. After careful consideration, we feel that it has merit but does not fully meet PLOS ONE’s publication criteria as it currently stands. Therefore, we invite you to submit a revised version of the manuscript that addresses the points raised during the review process.

We look forward to receiving your revised manuscript.

Kind regards,

Kiyoshi Sanada, PhD

Academic Editor

PLOS ONE

Journal Requirements:

 [no]. 

4.Thank you for stating the following in the Acknowledgments Section of your manuscript: 

[We thank Ms. Winnie for her English writing and correction

We also thank the Teh-Tzer Study Group for Human Medical Research Foundation for the support. The funders had no role in the study design, data collection, analysis, decision to publish, or preparation of the manuscript.]

 [no]

[no]. 

7. PLOS requires an ORCID iD for the corresponding author in Editorial Manager on papers submitted after December 6th, 2016. Please ensure that you have an ORCID iD and that it is validated in Editorial Manager. To do this, go to ‘Update my Information’ (in the upper left-hand corner of the main menu), and click on the Fetch/Validate link next to the ORCID field. This will take you to the ORCID site and allow you to create a new iD or authenticate a pre-existing iD in Editorial Manager. Please see the following video for instructions on linking an ORCID iD to your Editorial Manager account: https://www.youtube.com/watch?v=_xcclfuvtxQ

Reviewers' comments:

Reviewer's Responses to Questions

**Comments to the Author**

1. Is the manuscript technically sound, and do the data support the conclusions?

Reviewer #1: No

Reviewer #2: Yes

2. Has the statistical analysis been performed appropriately and rigorously? 

Reviewer #1: I Don't Know

Reviewer #2: No

3. Have the authors made all data underlying the findings in their manuscript fully available?

Reviewer #1: Yes

Reviewer #2: Yes

4. Is the manuscript presented in an intelligible fashion and written in standard English?

Reviewer #1: Yes

Reviewer #2: No

5. Review Comments to the Author

Reviewer #1: Major comments

I believe the topic of this paper is rather timely and appropriate, as this area of research has been expanding in recent years. However, the justification for the experimental protocol and discussion are very weak.

Table 6

Materials and Methods

Interobserver and intraobserver ICC is also needed in all measurements (Grip strength, body waist circumference, etc.).

Additionally, the Grip strength was very important point in this study, the authors should measured a few times. The test-retest reliability (ICC, SEM and minimal difference) is needed.

Minor comments

Limitation

Although only one limitation was proposed, it was written as some limitations.

It needs to be revised.

Table 1

As the data in Table 2 women, all data in the upper half require “±”.

Reviewer #2: [Peer Review Summary]

All the paragraphs are interesting, but many parts are not fully explained, making them difficult to understand. There are issues with the way the sentences are arranged. Therefore, to understand the claims of this study accurately, to emphasize the novelty and social significance, the structure needs to be revised. For this, you can create a better flow by replacing the context and summarizing what you want to emphasize in each paragraph. I suggest you categorize what you are saying in each section and reconsider the order in which you explain it. There are also grammatical errors such as spelling, definite articles, singular, and plural. Please check it.

---------------

Minor＝None

Major=★

---------------　

[Abstract ]

1.Anthropometric is generally recognized as height, weight, body mass index (BMI), body circumference (waist, hips, extremities), sebaceous thickness. However, this study also describes the association between grip strength and medical diseases, so this should be added to the title or modified by changing the title.

★2.Background

The title gives a strong impression that this study focuses on the relationship between grip strength and anthropometric characteristics. However, in the background, the topics of sarcopenia and frailty are discussed. It seems to make it difficult for readers to understand the content. Therefore, if you would like to argue about sarcopenia and frailty, you had better consider as follows: GS and muscle strength, GS and sarcopenia or Frailty, and GS and anthropometric characteristics. These items should be discussed more from the results especially in the discussion section.

3.Design and Method

Please indicate the number of subjects, their age and gender.

4.Results

The number of participants and subject details should be described in Design and Method

[Introduction]

★1.This is the same as the comments in the abstract.

There was a strong impression that this study focuses on the relationship between grip strength and anthropometric characteristics. However, in the background, the topics of sarcopenia and frailty are discussed. It seems to make it difficult for readers to understand the content. Therefore, if you would like to argue about sarcopenia and frailty, you had better consider as follows: GS and muscle strength, GS and sarcopenia or Frailty, and GS and anthropometric characteristics. These items should be debited from the results especially in the discussion section.

2.There should be some kind of conjunction to connect the sentence with the previous sentence as follows: The relationships between sarcopenia, obesity, and central obesity are not well established. This sentence could start with ‘However,….’.

★3.Reference 9 is a research paper on central obesity, sarcopenia, and nutrition. This paper concludes that central obesity and sarcopenia were interrelated with nutritional status in the elderly. However, you state that it is not well established. I would like to know your thoughts or comments on this.

★4.The results and discussion are inadequate for the study; similar comments are in the abstract. It needs to be written more coherently. This study aimed to clearly state the epidemiological characteristics of sarcopenia and frailty, but the results also emphasize the relationship between GS and anthropometric parameters, so I think the overall sentence structure needs to be revised in the abstract.

[Materials and Methods]

1.Study Populations

Please indicate the number of subjects, their age and gender.

2.previous three weeks

Simple question, is there any evidence that it is 3 weeks?

★3.Questionnaire

Association between the results of the questionnaire and the results of the anthropometric measurements should be explained in the Discussion. Everything that you have experimented with must be discussed.

4.There is insufficient information about the equipment used for the measurement. Only the grip strength tester is listed. Please provide information on all devices.

★5.GS measurement

Reference 22, this study examined the relationship between elbow joint angle and grip strength in subjects aged 20-57. Although grip strength is generally measured with the elbow extended. It's unclear about the purpose of measuring it in a sitting position and with the only dominant hand. An explanation or intention for applying this method to the elderly is needed.

6.Statistical methods

I think the grouping cutoff (e.g. normal, weak) should be indicated in the text as well. Also please check the other groupings.

[Results]

★1.I don't think it is necessary here to restate subject details such as age; in Results, you should simply state the results of your analysis.

2.Need to include the results of medical disease risk from blood samples.

3.In the sentence as follows; “compared with other anthropometric measures, only WHR showed a negative correlation, but the difference was without or borderline statistical significance in both sexes (r = -0.013, p = 0.613 in males and r = -0.047, p = 0.030 in females)”. For females, the significance is lower, but it is 0.03. I think you need to modify the explanation method.

4.The reason for Model 1 and Model 2 classification is unclear. I do not understand on what basis the models were classified. I think it would be better to indicate whether the classification was based on controllable factors such as congenital or acquired factors.

[Discussion]

★1.Insufficient consideration of measurement results. As commented in the questionnaire section, everything that has been experimented with should be discussed.

2.In the sentence as follows; “In the current ageing society, sarcopenia and physical disability among the elderly population is becoming increasingly important”. What is important? How is it important? Please explain.

3.In the sentence as follows; “although no obvious and clear mechanism has been discovered”. It is unclear from this what has not been clarified.

★4.In the sentence, as follows; “This study was the first large community-observed prospective study to investigate the relationship between anthropometric characteristics and GS among individuals older than 65 years in Taiwan”. A poor match between title and research content. This is the same as the comments in the abstract.

5.In the sentence as follows; “This result can be explained by the fact that taller individuals also have longer bones, which gives them greater GS”, Reference 32 is a twin study. It shows that GS changes with growth, but I don't think it says that " long bones have stronger GS". It is good to list the literature, but I think this sentence should be deleted or changed to another way of explanation.

★6.In the sentence, as follows; “This study applied simple personal anthropometric profile〜”. What you have shown in this paragraph is a unique result. If you want to emphasize the novelty here, you should include more discussion about this.

[Conclusion]

1.In the sentence as follows; “To conclude, our study showed that GS in the elderly Taiwanese population was similar to reports on GS in other Asian groups and was weaker than the GS of Western populations”. The term "Taiwanese elderly" is very broad, so I think it would be better to change the wording. Also, which report is being compared to be lower than the Western population?

★2.About the sentence as follows; “muscle weakness among elderly population in Taiwan.” It said, it predicts muscle weakness, but in your discussion, you say, "GS is considered an essential parameter in clinical settings to assess multiple risk factors such as cardiovascular and physical disorders." 1 be better to describe not only the prediction of muscle weakness but also the relationship with other parameters.

[Table1,2,3,4]

1.I think it would be better to show the cutoff values for Normal and Weak, and I have made the same comment in Statistical methods.

2.In Tables 2 and 4, there are no asterisks in the tables for p-values. 1A and 1B have asterisks, and there is a supplementary note below the tables saying **p<0.01, *p<0.05. Therefore, I think it should be standardized for all tables.

6. PLOS authors have the option to publish the peer review history of their article (what does this mean?). If published, this will include your full peer review and any attached files.

Reviewer #1: **Yes: **Tomohiro Yasuda, PhD

Reviewer #2: No

---

## [Decision Letter · Decision Letter 1]

25 Oct 2021

PONE-D-21-24416R1Association between grip strength and anthropometric characteristics in the community-dwelling elderly population in TaiwanPLOS ONE

Dear Dr. CHU,

Thank you for submitting your manuscript to PLOS ONE. After careful consideration, we feel that it has merit but does not fully meet PLOS ONE’s publication criteria as it currently stands. Therefore, we invite you to submit a revised version of the manuscript that addresses the points raised during the review process.

We look forward to receiving your revised manuscript.

Kind regards,

Kiyoshi Sanada, PhD

Academic Editor

PLOS ONE

Journal Requirements:

Reviewers' comments:

Reviewer's Responses to Questions

**Comments to the Author**

1. If the authors have adequately addressed your comments raised in a previous round of review and you feel that this manuscript is now acceptable for publication, you may indicate that here to bypass the “Comments to the Author” section, enter your conflict of interest statement in the “Confidential to Editor” section, and submit your "Accept" recommendation.

Reviewer #1: (No Response)

Reviewer #2: (No Response)

2. Is the manuscript technically sound, and do the data support the conclusions?

Reviewer #1: Partly

Reviewer #2: Yes

3. Has the statistical analysis been performed appropriately and rigorously? 

Reviewer #1: I Don't Know

Reviewer #2: Yes

4. Have the authors made all data underlying the findings in their manuscript fully available?

Reviewer #1: Yes

Reviewer #2: Yes

5. Is the manuscript presented in an intelligible fashion and written in standard English?

Reviewer #1: Yes

Reviewer #2: Yes

6. Review Comments to the Author

Reviewer #1: >Interobserver and intraobserver ICC is also needed in all measurements (Grip

strength, body waist circumference, etc.).

Additionally, the Grip strength was very important point in this study, the

authors should measured a few times. The test-retest reliability (ICC, SEM and

minimal difference) is needed.

Thank you for your valuable comments. We did our best to train our staff to

reach the inter-observer variation less than 5% and intra-observer variation

around 3-5%.

It should be described in the Materials and Methods and shown to the reader, not just the reviewer.

It is necessary for the reader to judge whether the measurement was performed under appropriate conditions.

Reviewer #2: In my last review, I had asked for a correction regarding the p-value.

In that case, for example, in Table 2, only one part was marked with an asterisk. Therefore, it was suggested that it would be better to unify them.

In this case, using Table 2 as an example, all women are marked with a symbol, while men are not.

The intention of the last review was that unification meant that if you wanted to add a symbol, you would put it on everything, and if not, you would remove it all.

This is a decision based on your thoughts.

7. PLOS authors have the option to publish the peer review history of their article (what does this mean?). If published, this will include your full peer review and any attached files.

Reviewer #1: **Yes: **Tomohiro Yasuda

Reviewer #2: No

---

## [Author Response · Author response to Decision Letter 1]

27 Oct 2021

We have revised and modified our update manuscript to fit reviewers’ comments.

---

## [Decision Letter · Decision Letter 2]

17 Nov 2021

Association between grip strength and anthropometric characteristics in the community-dwelling elderly population in Taiwan

PONE-D-21-24416R2

Dear Dr. CHU,

We’re pleased to inform you that your manuscript has been judged scientifically suitable for publication and will be formally accepted for publication once it meets all outstanding technical requirements.

Kind regards,

Kiyoshi Sanada, PhD

Academic Editor

PLOS ONE

Additional Editor Comments (optional):

Reviewers' comments:

Reviewer's Responses to Questions

**Comments to the Author**

1. If the authors have adequately addressed your comments raised in a previous round of review and you feel that this manuscript is now acceptable for publication, you may indicate that here to bypass the “Comments to the Author” section, enter your conflict of interest statement in the “Confidential to Editor” section, and submit your "Accept" recommendation.

Reviewer #1: All comments have been addressed

Reviewer #2: (No Response)

2. Is the manuscript technically sound, and do the data support the conclusions?

Reviewer #1: Yes

Reviewer #2: Yes

3. Has the statistical analysis been performed appropriately and rigorously? 

Reviewer #1: Yes

Reviewer #2: Yes

4. Have the authors made all data underlying the findings in their manuscript fully available?

Reviewer #1: Yes

Reviewer #2: Yes

5. Is the manuscript presented in an intelligible fashion and written in standard English?

Reviewer #1: Yes

Reviewer #2: Yes

6. Review Comments to the Author

Reviewer #1: Thank you for the revision.

I have no additional comments for the author.

I feel that this manuscript is now acceptable for publication.

Reviewer #2: (No Response)

7. PLOS authors have the option to publish the peer review history of their article (what does this mean?). If published, this will include your full peer review and any attached files.

Reviewer #1: **Yes: **Tomohiro Yasuda, PhD

Reviewer #2: No

---

## [Editor Report · Acceptance letter]

7 Dec 2021

PONE-D-21-24416R2 

Association between grip strength and anthropometric characteristics in the community-dwelling elderly population in Taiwan 

Dear Dr. Chu:

I'm pleased to inform you that your manuscript has been deemed suitable for publication in PLOS ONE. Congratulations! Your manuscript is now with our production department. 

Kind regards, 

on behalf of

Dr. Kiyoshi Sanada 

Academic Editor

PLOS ONE